# Adaptation and validation of HIV/AIDS module for university students in Larkana, Sindh, Pakistan

**Sarmad Jamal Siddiqi[1], Rosnah Sutan[2], Mariam Ashraf[3‡]*, Sadia Ajaib[4‡],
Zaleha Md Isa[5‡], Shabnam Naz[6], Ghulam Abbas Qadri[7], Rizwan Ali Tunio[8]**

1 Faculty of Community Medicine & Public Health Science, SMBB Medical University Larkana, Larkana,
Pakistan, 2 Public Health (Family Health), Population Health Research UKM, Department of Public Health
Medicine, Medical Faculty, Universiti Kebangsaan Malaysia (UKM), Kuala Lumpur, Malaysia, 3 School
of Public Health, Health Services Academy, Islamabad, Pakistan, 4 Department of Community Medicine,
Shifa Tameer-e-Millat University, Islamabad, Pakistan, 5 Department of Public Health Medicine, Faculty of
Medicine, Universiti Kebangsaan Malaysia, Cheras, Kuala Lumpur, Malaysia, 6 SMBB Medical University,
Larkana, Pakistan, 7 SMBB Medical University, Larkana, Pakistan, 8 Chandka Medical College, SMBBM
University Larkana, Larkana, Pakistan

‡ These authors contributed equally to this work.
* maryamashraf@hotmail.com

org/10.1371/journal.pgph.0006159

University South Tehran Branch, IRAN,
ISLAMIC REPUBLIC OF

**Peer Review History:** PLOS recognizes the
benefits of transparency in the peer review
process; therefore, we enable the publication
of all of the content of peer review and
author responses alongside final, published
articles. The editorial history of this article is
available here: https://doi.org/10.1371/journal.
pgph.0006159

## Abstract

University students in Pakistan frequently possess limited and inaccurate under-
standing of HIV/AIDS, including misconceptions about how it spreads, immunity, and
treatment. This knowledge gap increases their vulnerability, as young adults are a
high-risk group for new infections. The study focused on adapting and validating a
culturally suitable HIV/AIDS educational module specifically for university students
in Larkana, Sindh, Pakistan. The validation process consisted of three stages: (i)
adapting an existing HIV/AIDS educational package; (ii) conducting qualitative
research with religious scholars to confirm its cultural and contextual relevance; and
(iii) performing face and content validation with university students and public health
experts. Forty-eight students evaluated face validity using a binary scale, while six
subject-matter experts and five religious scholars assessed content validity via a
4-point Likert scale. Fleiss' Kappa Index (FKI) was employed to measure inter-rater
agreement, and Content Validity Indices (CVI) were calculated to assess overall
content validity. High agreement levels were observed: students (Kappa = 0.80) and
experts (Kappa = 0.82) showed strong consensus on the clarity and relevance of
the module content. The Scale-level Content Validity Index (S-CVI/Ave) was above
0.90 for both groups, 0.95 for experts, and 0.94 for religious scholars, exceeding
recommended standards. These results suggest that the module is both acceptable
and effective in delivering HIV/AIDS information in a culturally sensitive way. This
study demonstrates that a well-adapted HIV/AIDS educational module can achieve
high validity when tailored to local cultural and religious contexts. The module shows
promise as a model that can be replicated for HIV prevention and health education
programmes among youth in similar low- and middle-income settings.

**Data availability statement:** All data required to replicate the findings of this study are available within the manuscript and its supporting information files, including the raw agreement ratings and content validity index calculations presented in Tables 1–4.

**Funding:** The authors received no specific funding for this work.

**Competing interests:** The authors have declared that no competing interests exist.

## Introduction

The prevalence of HIV has drastically expanded around the world. More than 60 million individuals worldwide are living with or have been affected by HIV, including over 20 million people who have lost their lives to this disease, and more than ten people become newly infected every minute. In developing countries, more than half of new HIV infections occur among young people under the age of 25 years, including Pakistan, where 60% of new infections occur among young people between the ages of 19 and 29 years. In 2021, there were approximately 38.4 million people worldwide who tested positive for HIV, with a range of 33.9 million to 43.8 million individuals [1]. In the Asia-Pacific region, young people aged 15–24 years account for approximately 15% of new HIV infections, with significant knowledge gaps persisting across South and Southeast Asian countries [2]. Studies from neighboring countries have documented poor HIV/AIDS awareness among young adults, with misconceptions about transmission routes and prevention methods remaining prevalent [3],

The population of young adults is increasing rapidly around the world, and risky sexual practices are becoming more common among this group [4]. Education regarding sexual and reproductive health has remained limited and a controversial topic in numerous countries globally [5]. Globally, 0.7% (0.6-0.8%) of adults between the ages of 15 and 49 are thought to be HIV positive, while the epidemic's toll varies greatly between nations and areas. With approximately 1 in every 25 adults (3.2%) living with HIV and making up more than two-thirds of all HIV-positive individuals globally, the WHO African Region continues to be the most severely afflicted [6].

In Pakistan, the demographic increase in the youth population, decreased level of literacy, decreased awareness about sexually transmitted infections, and weak health indicators are important risk factors for increasing vulnerability of youngsters towards sexually transmitted diseases [7,8]. Beyond knowledge gaps, people living with HIV/AIDS in Pakistan face significant psychosocial challenges, including anxiety and impaired social functioning, which further underscore the need for comprehensive, stigma-reducing educational interventions targeting at-risk youth populations[9]. Despite these facts, the public in Pakistan has not yet acknowledged that sexually transmitted diseases have anything to do with them. People consider HIV/AIDS a highly shameful disease, especially in rural areas of Pakistan. Awareness towards HIV/AIDS among the general public in Pakistan is restricted. A survey showed that school teachers in Pakistan thought that HIV was irrelevant in Pakistani cultural settings [9].

Research shows that Pakistani university students typically lack a precise and thorough understanding of HIV/AIDS, and misconceptions about immunity, treatment, and transmission are still prevalent. The results of the study emphasize how crucial it is for successful educational interventions to close these gaps and advance truthful awareness, particularly among those studying subjects other than medicine [10].

Therefore, it is necessary to educate university students in Larkana city so that they have enough knowledge towards HIV/AIDS, as its prevalence is increasing, so that they can protect themselves and prevent the disease, as they are the most

affected group of society. Additionally, the role of religious leaders is significant in raising awareness and educating young adults [11]. Therefore, religious scholars have also been included in this research.

## Materials and methods

### Ethics statement

The research study was approved by the Human Research Ethics Committee of Universiti Kebangsaan Malaysia, reference number UKM PPI/111/8/JEP-2021–853 and the Shaheed Mohtarma Benazir Bhutto Medical University (SMBBMU), Larkana, Sindh, number SMBBMU/OFF ERC/187. Ethics approval from both institutions was required as the first author was enrolled as a PhD student at Universiti Kebangsaan Malaysia during study design and data collection; UKM institutional policy mandates ethics approval for all student research regardless of geographical location, while SMBBMU approval ensured compliance with local ethical standards. Formal consent was obtained from all the participants, ensuring anonymity and confidentiality following the Declaration of Helsinki.

### Study setting and site selection

This validation study was conducted at Shaheed Mohtarma Benazir Bhutto Medical University (SMBBMU), Larkana, Sindh, Pakistan, between June and August 2022. SMBBMU is a public-sector medical university established in 2008, serving the Larkana region and surrounding districts of northern Sindh province.

The university was selected for several reasons: (1) Larkana district has reported higher rates of HIV prevalence compared to other districts in Sindh province; (2) the university serves a diverse student population from both urban and rural backgrounds; (3) limited formal HIV/AIDS education exists in the university curriculum outside of medical programs; and (4) institutional support and infrastructure were available to conduct the study.

QUEST (Quaid-e-Awam University of Engineering, Science, and Technology) is a separate public-sector engineering university also located in Larkana, established in 2012, offering programs in engineering, computer science, and technology. Students from QUEST were included to ensure representation from non-medical disciplines.

Of the 48 students who participated in face validity assessment: 24 students (50%) were from SMBBMU medical and allied health programs, and 24 students (50%) were from QUEST engineering and technology programs. This balanced distribution ensured representation from both biomedical and non-biomedical educational backgrounds.

### Sample size determination

For face validity assessment, the sample size was determined following the recommendations of Hertzog (2008) and Taherdoost (2016), which suggest that 10–20 participants per item category is adequate for preliminary face validity testing of educational materials. Given our module contained 8 major content domains, a minimum of 40–48 participants was considered appropriate.

For content validity, we followed Lynn's (1986) [12] recommendation of 6–10 subject-matter experts. Our panel of 6 public health experts and 5 religious scholars (total n = 11) exceeded this minimum requirement.

Face and content validity studies focus on the quality of expert judgment and comprehensiveness of content coverage rather than population representativeness. This methodological approach is consistent with established validation protocols in health education research [13].

### Study design and duration

The study was conducted in three phases: Phase I involved the adaptation and development of an HIV educational module, followed by Phase II, which included qualitative research, and finally, Phase III, which involved content and face validation. These phases of the study were conducted from June 2022 to January 2023.

## Data collection and statistical analysis

Data collection was conducted over 8 weeks (June-August 2022). Face validity data were collected through structured individual interviews, while content validity assessments were completed by experts using standardized forms. All responses were coded and entered into Microsoft Excel 2019.

Statistical analyses were performed using IBM SPSS Statistics Version 26.0 and R Software Version 4.2.1. Descriptive statistics (frequencies and percentages) were calculated for demographic variables.

For face validity, Fleiss' Kappa was calculated to measure inter-rater reliability. For content validity, responses on the 4-point Likert scale were dichotomized (3–4 = relevant, 1–2 = not relevant) following Lynn (1986) and Polit & Beck (2006). Item-level Content Validity Index (I-CVI) and Scale-level Content Validity Index (S-CVI/Ave) were computed.

Fleiss' Kappa values were interpreted according to Landis and Koch (1977): < 0.00 (poor), 0.00-0.20 (slight), 0.21-0.40 (fair), 0.41-0.60 (moderate), 0.61-0.80 (substantial), 0.81-1.00 (almost perfect). I-CVI values ≥0.78 and S-CVI/Ave ≥ 0.90 were considered acceptable.

## Phase I: Adaptation and development of health educational module

A literature review explored existing publications, educational modules, and awareness packages regarding HIV/AIDS. Electronic sources and websites were involved in exploring the data, including PubMed, the Cochrane Library database, the Web of Science, Google Scholar, the UNAIDS website, and the UNESCO website.

The HIV/AIDS handout of the Sexual and Reproductive Health Training Manual for Young People, developed by the German Foundation for World Population in 2006 was adopted [14].

Most religious scholars did not understand English, so the selected package was translated into Urdu, Pakistan's national language. Two experienced linguistics lecturers, one with a master's degree in English and the other with a master's degree in Urdu, were hired to translate into Urdu and back, ensuring the meanings remained the same. The Urdu version of the module was then finalized.

## Phase II: Qualitative research study with religious scholars

To assess the views of religious scholars regarding the education module on HIV/AIDS for university students in Pakistan, a qualitative study was carried out, involving in-depth interviews with religious scholars in Larkana, Pakistan. From a religious perspective, the HIV educational package was improved by incorporating suggestions from religious scholars to ensure it was religiously acceptable in Pakistan for the benefit of youth and university students.

A structured questionnaire guide was developed, incorporating 8 open-ended questions. The questions addressed all the topics and issues included in the package and examined the perspectives of religious scholars on religion's role in providing health education and awareness to prevent HIV/AIDS among university students and youth. The questionnaire was created in accordance with the guidelines of Jordan et al [15].

Religious scholars holding at least a "Shahadatul Alia fil Uloom Arabia Wal Islamia," equivalent to a Bachelor's degree in Arts in Islamic Studies from a Madrasa in Pakistan, and who agreed to participate in the research, were interviewed. A total of five religious scholars were interviewed, achieving data saturation. The in-depth interviews were conducted in Urdu using the Urdu translation of the Package. Comprehensive documentation of these interviews was maintained. All topics covered in the HIV/AIDS educational module were thoroughly explored during the interviews with the religious scholars.

To compile a comprehensive report of the discussion, using the participants' own words, a document was prepared that accurately reflected the debate. The researcher then reviewed the completed transcriptions to ensure that all discussed points were thoroughly captured. It is noteworthy that each interview was documented, reviewed by the researcher, and transcribed within twenty-four hours, capturing the precise words and phrases. Key words, phrases, and significant quotes

illustrating specific patterns were recorded, and all the interviews were subsequently translated into English by two experienced linguistics lecturers: one specialising in Urdu and the other in English.

The interviews took place in a location convenient for religious scholars and lasted between 45 and 60 minutes for each session. The questions were open-ended, allowing participants to express their ideas regarding the content, design, and format of the module they preferred.

Thematic analysis was conducted manually. The principal investigator who carried out the interviews read the transcripts and identified significant findings related to the needs for the module's content and design. Codes were identified and categorised into sub-themes and themes. These codes, sub-themes, and themes were discussed among the research team members. Improvements were made until a consensus was reached.

### Phase III: Content and face validation

The content validity of an instrument is about how well it represents the idea and construct under consideration. On the other hand, the face validity of an instrument is about its apparent suitability and relevance to the concept, idea, or construct. Thus, face validity is a subjective assessment of an instrument's appearance, presentation, relevance, reasonableness, and clarity. On the other hand, content validity is about the representativeness and adequacy of the content of the questions or items constructed to measure a phenomenon [16].

To determine the face and content validity of the HIV/AIDS module, the module was assessed among the targeted students at an engineering university (QUEST) and a medical university (SMBBMU). Whereas the Face and content validity were determined through the views and input of the relevant experts (for English version) of the module, and Content validity was determined by experts to assess the content of the module.

The validation process of the HIV/AIDS module was administered systematically. As a first step, two validation tools were developed to get the qualitative insights of the experts and quantitative feedback from the students and experts on items of different module domains. The validation tools attempted to assess the difficulty level, clarity, interpretation, comprehensiveness, and understandability of the module. Based on the scope and content of the module, 8 domains were identified that were then tested on 06 items seeking students' and experts' agreement on binary options (Yes or No) related to words, spellings, grammar, sentences, and overall structure of the section for face validity. The agreement of the religious scholars and experts was sought for content validity on a 4-point Likert scale, based on relevance, simplicity, clarity, delivery, and understandability (see validation tools) of the module. The domains tested through the validation tool were:

1. Definition and explanation of HIV/AIDS

2. HIV infection and the human immune system

3. Transition from HIV to AIDS

4. HIV transmission and prevention

5. Voluntary counselling and testing

6. Medical Check-ups of HIV/AIDS

7. HIV/AIDS interventions

8. Helping HIV positive and AIDS patients

A total of 48 students were selected and randomly assigned to assess the face validity of the manual. The sample was distributed proportionally to the previously defined strata, which represent the types of universities. A total of six experts were chosen using a convenient sampling method to evaluate the content validity. This group of experts included two

Public Health Faculty members with expertise in Health Promotion, both holding PhDs in Public Health. Additionally, two consultant physicians were selected as clinical experts; both are FCPS in Medicine. Furthermore, two officers designated as monitoring and evaluation officers working on the UNDP-GF Project for the prevention and control of HIV/AIDS were also selected as experts in this area. The content validity of the Urdu-language translated version of the module was assessed among the same five religious scholars selected for the qualitative phase of the study from the chosen Madrasa to evaluate the translation.

## Face validity

For face validity of the manual, a dichotomous scale was used with binary options of 'Yes' and 'No', which indicated favourable and unfavourable responses, respectively. Favourable items mean that the item was objectively structured and relevant to the intended information. Those criteria are:

i. Appropriateness of grammar

ii. Clarity of items

iii. Correct spellings

iv. Correct structure of sentences

v. Suitability of font size

vi. Structure and construction of the section

The assessment tool for face validity of the manual was distributed among the students and clear instructions were given related to the assessment criteria and recording of responses and the students attempted to assess the difficulty level of the module, clarity and interpretation of items. Comprehensibility of the terminologies and words used in the instrument and same process was used to assess face validity by experts as well. Although the experts played a vital role in the content validation, the review by a sample of target subjects (students) added to the validation of the face as an essential component. They were asked to record their response as 'Yes' or 'No'.

The responses collected from the students and experts were analysed separately using Fleiss' Kappa Index (FKI), introduced by Fleiss, to assess the face validity of an instrument. Thus, FKI measures the degree of agreement with the theme or unit of analysis of the constructs studied and validated by two or more experts who agree on similar or identical ratings. A face validity measure of Kappa equal to +1 indicates perfect agreement among rating experts, while 1' signifies perfect disagreement If Kappa takes the value 0, it suggests no correlation between the experts' ratings and that any agreement or disagreement occurs purely by chance [17].

## Face validity by students

Table 1 shows the face validity evaluation by 48 students and 6 experts, who assessed various criteria within the HIV/AIDS education package. Criteria such as grammatical appropriateness, clarity of items, and correct spelling received positive feedback from the evaluators.

Overall, the data indicate that most students and experts viewed the face validity of the HIV/AIDS education package positively. Sections related to HIV/AIDS interventions received the highest approval, while sections such as Voluntary Counselling and Testing garnered slightly lower approval ratings but were still generally well-received.

## Content validation

For content validity, the module was distributed to six Subject Matter Experts (via email) and five Religious Scholars (in person) for the Urdu translated version. The religious scholars were also briefed about the content and objectivity of the module and were requested to consider it from local and religious perspectives. The experts received detailed instructions regarding

**Table 1. Students and experts module face validity (Students n = 48, Experts n = 6).**

| Evaluation Criteria | Students | Experts |
|---|---|---|
| | **Agree** | **Agree** |
| **1-Definition and explanation of HIV/AIDS** | **80%** | **92%** |
| Appropriateness of grammar | 36 | 5 |
| Clarity of items | 39 | 6 |
| Correct spelling of words | 38 | 5 |
| Correct structure of sentences | 37 | 6 |
| Suitability of font size | 40 | 6 |
| Structure and construction of the section | 41 | 5 |
| **2. HIV Infection and Human Immune System** | **83%** | **100%** |
| Appropriateness of grammar | 42 | 6 |
| Clarity of items | 40 | 6 |
| Correct spelling of words | 38 | 6 |
| Correct structure of sentences | 40 | 6 |
| Suitability of font size | 42 | 6 |
| Structure and construction of the section | 37 | 6 |
| **3. Transition from HIV to AIDS** | **83%** | **92%** |
| Appropriateness of grammar | 44 | 6 |
| Clarity of items | 42 | 6 |
| Correct spelling of words | 39 | 6 |
| Correct structure of sentences | 35 | 4 |
| Suitability of font size | 38 | 6 |
| Structure and construction of the section | 41 | 5 |
| **4. HIV transmission and prevention** | **80%** | **100%** |
| Appropriateness of grammar | 42 | 5 |
| Clarity of items | 42 | 6 |
| Correct spelling of words | 39 | 6 |
| Correct structure of sentences | 42 | 6 |
| Suitability of font size | 38 | 6 |
| Structure and construction of the section | 44 | 6 |
| **5. Voluntary counselling and testing** | **77%** | **97%** |
| Appropriateness of grammar | 43 | 6 |
| Clarity of items | 37 | 6 |
| Correct spelling of words | 37 | 6 |
| Correct structure of sentences | 34 | 6 |
| Suitability of font size | 37 | 6 |
| Structure and construction of the section | 34 | 5 |
| **6. Medical Check-ups of HIV/AIDS** | **82%** | **94%** |
| Appropriateness of grammar | 40 | 6 |
| Clarity of items | 35 | 5 |
| Correct spelling of words | 38 | 5 |
| Correct structure of sentences | 40 | 6 |
| Suitability of font size | 39 | 6 |
| Structure and construction of the section | 43 | 6 |
| **7. HIV/AIDS Interventions** | **89%** | **94%** |
| Appropriateness of grammar | 41 | 4 |
| Clarity of items | 43 | 6 |

*(Continued)*

**Table 1.** (Continued)

| Evaluation Criteria | Students | Experts |
|---|---|---|
| | **Agree** | **Agree** |
| Correct spelling of words | 41 | 6 |
| Correct structure of sentences | 43 | 6 |
| Suitability of font size | 45 | 6 |
| Structure and construction of the section | 42 | 6 |
| **8. Helping out HIV+ve and AIDS patients** | **82%** | **100%** |
| Appropriateness of grammar | 36 | 6 |
| Clarity of items | 39 | 6 |
| Correct spelling of words | 48 | 6 |
| Correct structure of sentences | 39 | 6 |
| Suitability of font size | 35 | 6 |
| Structure and construction of the section | 38 | 6 |

the assessment criteria and scoring to be used in the content validation tool. In addition to scoring, the experts were asked to identify areas for improvement and provide suggestions. This approach facilitated the collection of qualitative insights from the experts on the content of the module, helping to enhance the language, examples, and references used in the health education module. The experts were instructed to rate their responses on a four-point Likert scale (Disagree, Partially Agree, Agree, and Completely Agree) applied to six items in each domain of the manual, covering a total of eight domains.

The Content Validation Index (CVI) was used as an assessment tool, as many researchers do, to evaluate the content validity of tools. For this purpose, the study calculated the content validation indices recommended by Lynn (1986) [12], and Polit & Beck [13]. Before calculating the CVI, the agreement ratings of 1.2 and 3.4 were noted as '0' and '1', respectively. Then, different calculations were made for various indices as follows:

a) Experts/Student agreement: Linear addition of rating by all experts and students

b) Universal agreement (UA): The item receiving a '1' score from all experts is assigned an overall score of '1'.

c) Item-level content validity index (I-CVI): experts in agreement/total number of experts

d) Scale-level content validity index (S-CVI) based on I-CVI: average of I-CVI or I-CVI/total items.

e) Scale-level content validity index (S-CVI) based on the average of proportion relevance.

f) Scale-level content validity index (S-CVI) based on UA: I-CVI/UA.

The detailed calculation of CVIs is presented in Tables 2–4 outlining the calculation of scholars' and experts' agreement, universal agreement (UA), item-level content validity index (I-CVI), scale-level content validity index based on I-CVI (S-CVI), and S-CVI based on UA. The calculated values of S-CVI/Ave (0.95) and S-CVI/UA (0.81) exceeded the satisfactory level (0.70), indicating satisfactory content validity. Similarly, the calculated values of S-CVI/Ave and S-CVI/UA were 0.94 and 0.77, respectively.

## Discussion

The validation of the HIV/AIDS health education module for university students in Larkana, Pakistan, provides evidence that culturally adapted and contextually relevant interventions can effectively address knowledge gaps among young adults. The strong agreement observed in face and content validity measures demonstrates that the module is both scientifically sound and acceptable within the local socio-cultural context.

**Table 2. Content Validity Index.**

| Indices<br><br>Experts/Scholars | S-CVI based on I-CVI<br>S-CVI/Ave = sum of I-CVI/total items | S-CVI based on UA<br>S-CVI/UA = sum of I-CVI/UA |
| --- | --- | --- |
| Subject Matter Experts | 0.95 | 0.81 |
| Religious Scholars | 0.94 | 0.77 |

Our findings align with previous research emphasising the importance of culturally sensitive health education interventions in low- and middle-income countries (LMICs), primarily where stigma and misconceptions about HIV/AIDS persist [18]. Similar studies in South Asia and sub-Saharan Africa have shown that incorporating local cultural and religious perspectives into HIV education enhances the acceptability and uptake of prevention messages [19,20]

A key strength of this study was the involvement of religious scholars during the adaptation process. Their input ensured that the module addressed misconceptions while remaining aligned with cultural and religious values. This participatory approach aligns with global calls to engage community and faith leaders in HIV prevention and health promotion [16]. The rigorous validation methodology employed in this study mirrors approaches used in other recent health education intervention studies conducted in Pakistani educational settings, demonstrating the feasibility and importance of systematically validating culturally adapted materials before implementation [21]. By bridging biomedical knowledge with sociocultural perspectives, the module demonstrates a practical model for improving HIV-related awareness in conservative settings.

The high content validity indices (S-CVI/Ave > 0.90) confirm that the adapted module meets international standards of health education tool development. Such rigorous validation is essential to ensure effectiveness and replicability. The strong agreement across both students and experts underscores the module's relevance for diverse university populations, including those studying non-medical disciplines. Such high agreement rates underscore the potential impact of the health education intervention package in raising awareness and preventing the spread of HIV/AIDS among university students in Pakistan. The findings align with previous research highlighting the importance of targeted health education interventions in regions with a high prevalence of HIV/AIDS. For instance, studies have demonstrated the effectiveness of well-designed interventions in improving knowledge and awareness among university students in South Asian countries [22]. The strong agreement observed in this study reinforces the potential effectiveness of such interventions in the Pakistani context.

From a global health perspective, this work contributes to the realisation of the Sustainable Development Goals (SDG 3: Ensure healthy lives and promote well-being for all at all ages, notably Target 3.3: ending the AIDS epidemic by 2030). By enhancing HIV/AIDS knowledge among youth—a group disproportionately affected by new infections—the intervention has potential for wider application across LMICs facing similar challenges. Furthermore, the approach aligns with the WHO's Global Health Sector Strategy on HIV, which highlights health literacy, stigma reduction, and community participation [23].

## Conclusion

In conclusion, the validation of the health education intervention package on HIV/AIDS among university students in Larkana, Pakistan, has yielded promising results, evidenced by a high level of agreement among students and experts. These findings align with previous research in the field, reinforcing the notion that well-designed health education interventions can play a crucial role in raising awareness and preventing the spread of HIV/AIDS among university students in Pakistan. This study contributes to the expanding body of knowledge in this area and underscores the importance of sustained efforts in health education and awareness campaigns. The strong content and face validity of the module indicate its potential effectiveness in the Pakistani context, offering a valuable tool for health education and intervention efforts in the region.

**Table 3. Content Validation Index Calculation – Experts' agreement on the item scale.**

| Item | Expert 1 | Expert 2 | Expert 3 | Expert 4 | Expert 5 | Expert 6 | Experts in Agreement | I-CVI | UA |
|------|----------|----------|----------|----------|----------|----------|----------------------|-------|-----|
| 1.1 | 1 | 1 | 1 | 1 | 0 | 1 | 5 | 0.8 | 0 |
| 1.2 | 1 | 1 | 1 | 1 | 1 | 1 | 6 | 1.0 | 1 |
| 1.3 | 1 | 0 | 0 | 1 | 1 | 1 | 4 | 0.7 | 0 |
| 1.4 | 1 | 1 | 1 | 1 | 1 | 1 | 6 | 1.0 | 1 |
| 1.5 | 1 | 1 | 1 | 1 | 1 | 1 | 6 | 1.0 | 1 |
| 1.6 | 1 | 1 | 1 | 1 | 1 | 1 | 6 | 1.0 | 1 |
| 2.1 | 1 | 1 | 1 | 1 | 1 | 1 | 6 | 1.0 | 1 |
| 2.2 | 1 | 0 | 0 | 1 | 1 | 1 | 4 | 0.7 | 0 |
| 2.3 | 1 | 1 | 1 | 1 | 0 | 1 | 5 | 0.8 | 0 |
| 2.4 | 0 | 1 | 0 | 1 | 0 | 0 | 2 | 0.3 | 0 |
| 2.5 | 1 | 1 | 1 | 1 | 1 | 1 | 6 | 1.0 | 1 |
| 2.6 | 1 | 1 | 1 | 1 | 1 | 1 | 6 | 1.0 | 1 |
| 3.1 | 1 | 1 | 1 | 1 | 1 | 1 | 6 | 1.0 | 1 |
| 3.2 | 1 | 1 | 1 | 1 | 1 | 1 | 6 | 1.0 | 1 |
| 3.3 | 1 | 1 | 1 | 1 | 1 | 1 | 6 | 1.0 | 1 |
| 3.4 | 1 | 1 | 1 | 1 | 1 | 1 | 6 | 1.0 | 1 |
| 3.5 | 1 | 0 | 1 | 1 | 1 | 1 | 5 | 0.8 | 0 |
| 3.6 | 1 | 1 | 1 | 1 | 1 | 1 | 6 | 1.0 | 1 |
| 4.1 | 1 | 1 | 1 | 1 | 1 | 1 | 6 | 1.0 | 1 |
| 4.2 | 1 | 1 | 1 | 1 | 1 | 1 | 6 | 1.0 | 1 |
| 4.3 | 1 | 1 | 1 | 1 | 1 | 1 | 6 | 1.0 | 1 |
| 4.4 | 1 | 1 | 1 | 1 | 1 | 1 | 6 | 1.0 | 1 |
| 4.5 | 1 | 1 | 1 | 1 | 1 | 1 | 6 | 1.0 | 1 |
| 4.6 | 0 | 1 | 1 | 1 | 1 | 1 | 5 | 0.8 | 0 |
| 5.1 | 1 | 1 | 1 | 1 | 1 | 1 | 6 | 1.0 | 1 |
| 5.2 | 1 | 1 | 1 | 1 | 1 | 1 | 6 | 1.0 | 1 |
| 5.3 | 1 | 1 | 1 | 1 | 1 | 1 | 6 | 1.0 | 1 |
| 5.4 | 1 | 1 | 1 | 1 | 1 | 1 | 6 | 1.0 | 1 |
| 5.5 | 1 | 1 | 1 | 1 | 1 | 1 | 6 | 1.0 | 1 |
| 5.6 | 1 | 1 | 1 | 1 | 1 | 1 | 6 | 1.0 | 1 |
| 6.1 | 1 | 1 | 1 | 1 | 1 | 1 | 6 | 1.0 | 1 |
| 6.2 | 1 | 1 | 1 | 1 | 1 | 0 | 5 | 0.8 | 0 |
| 6.3 | 1 | 1 | 1 | 1 | 1 | 1 | 6 | 1.0 | 1 |
| 6.4 | 1 | 1 | 0 | 1 | 1 | 1 | 5 | 0.8 | 0 |
| 6.5 | 1 | 1 | 1 | 1 | 1 | 1 | 6 | 1.0 | 1 |
| 6.6 | 1 | 1 | 1 | 1 | 1 | 1 | 6 | 1.0 | 1 |
| 7.1 | 1 | 1 | 1 | 1 | 1 | 1 | 6 | 1.0 | 1 |
| 7.2 | 1 | 1 | 1 | 1 | 1 | 1 | 6 | 1.0 | 1 |
| 7.3 | 1 | 1 | 1 | 1 | 1 | 1 | 6 | 1.0 | 1 |
| 7.4 | 1 | 1 | 1 | 1 | 1 | 1 | 6 | 1.0 | 1 |
| 7.5 | 1 | 1 | 1 | 1 | 1 | 1 | 6 | 1.0 | 1 |
| 7.6 | 1 | 1 | 1 | 1 | 1 | 1 | 6 | 1.0 | 1 |
| 8.1 | 1 | 1 | 1 | 1 | 1 | 1 | 6 | 1.0 | 1 |
| 8.2 | 1 | 1 | 1 | 1 | 1 | 1 | 6 | 1.0 | 1 |
| 8.3 | 1 | 1 | 1 | 1 | 1 | 1 | 6 | 1.0 | 1 |
| 8.4 | 1 | 1 | 1 | 1 | 1 | 1 | 6 | 1.0 | 1 |

*(Continued)*

**Table 3.** (Continued)

| Item | Expert 1 | Expert 2 | Expert 3 | Expert 4 | Expert 5 | Expert 6 | Experts in Agreement | I-CVI | UA |
|---|---|---|---|---|---|---|---|---|---|
| 8.5 | 1 | 1 | 1 | 1 | 1 | 1 | 6 | 1.0 | 1 |
| 8.6 | 1 | 1 | 1 | 1 | 1 | 1 | 6 | 1.0 | 1 |
| Agreement rating | 0.96 | 0.94 | 0.92 | 1.00 | 0.94 | 0.96 | S-IVC/Av | 0.95 (95%) | |
| | | | | | | | S-IVC/UA | | 0.81(81%) |

**Table 4.** Content Validation Index Calculation – Religious Scholars' agreement on the item scale.

| Item | Religious Scholar 1 | Religious Scholar 2 | Religious Scholar 3 | Religious Scholar 4 | Religious Scholar 5 | Experts in Agreement | I-CVI | UA |
|---|---|---|---|---|---|---|---|---|
| 1.1 | 1 | 1 | 0 | 1 | 1 | 4 | 0.8 | 0 |
| 1.2 | 1 | 1 | 1 | 1 | 1 | 5 | 1 | 1 |
| 1.3 | 1 | 0 | 1 | 1 | 1 | 4 | 0.8 | 0 |
| 1.4 | 1 | 1 | 1 | 1 | 1 | 5 | 1 | 1 |
| 1.5 | 1 | 1 | 1 | 1 | 1 | 5 | 1 | 1 |
| 1.6 | 0 | 1 | 1 | 1 | 0 | 3 | 0.6 | 0 |
| 2.1 | 1 | 1 | 1 | 1 | 1 | 5 | 1 | 1 |
| 2.2 | 1 | 1 | 0 | 1 | 1 | 4 | 0.8 | 0 |
| 2.3 | 1 | 1 | 1 | 1 | 0 | 4 | 0.8 | 0 |
| 2.4 | 0 | 1 | 1 | 1 | 1 | 4 | 0.8 | 0 |
| 2.5 | 1 | 1 | 1 | 1 | 1 | 5 | 1 | 1 |
| 2.6 | 1 | 1 | 1 | 1 | 1 | 5 | 1 | 1 |
| 3.1 | 1 | 1 | 1 | 1 | 1 | 5 | 1 | 1 |
| 3.2 | 1 | 1 | 1 | 1 | 1 | 5 | 1 | 1 |
| 3.3 | 1 | 1 | 1 | 1 | 1 | 5 | 1 | 1 |
| 3.4 | 1 | 1 | 1 | 1 | 1 | 5 | 1 | 1 |
| 3.5 | 1 | 0 | 1 | 1 | 1 | 4 | 0.8 | 0 |
| 3.6 | 1 | 1 | 1 | 1 | 1 | 5 | 1 | 1 |
| 4.1 | 1 | 1 | 1 | 1 | 1 | 5 | 1 | 1 |
| 4.2 | 1 | 1 | 1 | 1 | 1 | 5 | 1 | 1 |
| 4.3 | 1 | 1 | 1 | 1 | 1 | 5 | 1 | 1 |
| 4.4 | 1 | 1 | 1 | 1 | 1 | 5 | 1 | 1 |
| 4.5 | 1 | 1 | 1 | 1 | 1 | 5 | 1 | 1 |
| 4.6 | 1 | 1 | 1 | 1 | 1 | 5 | 1 | 1 |
| 5.1 | 1 | 1 | 1 | 1 | 1 | 5 | 1 | 1 |
| 5.2 | 1 | 1 | 1 | 1 | 1 | 5 | 1 | 1 |
| 5.3 | 1 | 1 | 1 | 1 | 1 | 5 | 1 | 1 |
| 5.4 | 1 | 1 | 1 | 1 | 1 | 5 | 1 | 1 |
| 5.5 | 1 | 1 | 1 | 1 | 1 | 5 | 1 | 1 |
| 5.6 | 1 | 1 | 1 | 1 | 1 | 5 | 1 | 1 |
| 6.1 | 0 | 1 | 1 | 1 | 1 | 4 | 0.8 | 0 |
| 6.2 | 0 | 1 | 1 | 1 | 0 | 3 | 0.6 | 0 |
| 6.3 | 1 | 1 | 1 | 1 | 1 | 5 | 1 | 1 |
| 6.4 | 1 | 1 | 0 | 1 | 1 | 4 | 0.8 | 0 |
| 6.5 | 1 | 1 | 1 | 1 | 1 | 5 | 1 | 1 |

*(Continued)*

**Table 4.** (Continued)

| Item | Religious Scholar 1 | Religious Scholar 2 | Religious Scholar 3 | Religious Scholar 4 | Religious Scholar 5 | Experts in Agreement | I-CVI | UA |
|---|---|---|---|---|---|---|---|---|
| 6.6 | 1 | 1 | 1 | 1 | 1 | 5 | 1 | 1 |
| 7.1 | 1 | 1 | 1 | 1 | 1 | 5 | 1 | 1 |
| 7.2 | 1 | 1 | 1 | 1 | 1 | 5 | 1 | 1 |
| 7.3 | 1 | 1 | 1 | 1 | 1 | 5 | 1 | 1 |
| 7.4 | 1 | 1 | 1 | 1 | 1 | 5 | 1 | 1 |
| 7.5 | 1 | 1 | 1 | 1 | 1 | 5 | 1 | 1 |
| 7.6 | 1 | 1 | 1 | 1 | 1 | 5 | 1 | 1 |
| 8.1 | 1 | 1 | 1 | 1 | 1 | 5 | 1 | 1 |
| 8.2 | 1 | 1 | 1 | 1 | 1 | 5 | 1 | 1 |
| 8.3 | 1 | 1 | 1 | 1 | 1 | 5 | 1 | 1 |
| 8.4 | 0 | 1 | 1 | 0 | 0 | 2 | 0.4 | 0 |
| 8.5 | 1 | 1 | 1 | 1 | 1 | 5 | 1 | 1 |
| 8.6 | 1 | 1 | 1 | 1 | 1 | 5 | 1 | 1 |
| Agreement rating | 0.90 | 0.96 | 0.94 | 0.98 | 0.92 | S-IVC/Av | 0.94 (94%) | |
| | | | | | | S-IVC/UA | | 0.77 (77%) |

## Limitations

Although the study shows promising results, some limitations need to be recognised. The validation was conducted with a relatively small group of students and experts from a single province, which may limit the extent to which the findings can be applied. Additionally, the study evaluated face and content validity but did not measure changes in knowledge or behaviour after implementing the module. Future research should aim to test the module's effectiveness in enhancing HIV-related knowledge, attitudes, and preventive practices across diverse educational and geographical contexts.

## Implications

The validated module offers a valuable, affordable, and reproducible tool for universities, NGOs, and public health organisations aiming to enhance HIV/AIDS education in culturally sensitive settings. Expanding such interventions, especially when tailored for regional languages and incorporated into curricula, can help close the ongoing knowledge gap and diminish stigma related to HIV in Pakistan and other LMICs.

## Author contributions

**Conceptualization:** Sarmad Jamal Siddiqi, Rosnah Sutan, Mariam Ashraf, Rizwan Ali Tunio.

**Data curation:** Sarmad Jamal Siddiqi.

**Formal analysis:** Sarmad Jamal Siddiqi, Rosnah Sutan, Mariam Ashraf, Sadia Ajaib, Zaleha Md Isa.

**Investigation:** Sarmad Jamal Siddiqi, Sadia Ajaib.

**Methodology:** Sarmad Jamal Siddiqi, Rosnah Sutan, Mariam Ashraf, Sadia Ajaib, Ghulam Abbas Qadri.

**Resources:** Sarmad Jamal Siddiqi.

**Supervision:** Rosnah Sutan, Mariam Ashraf, Zaleha Md Isa, Shabnam Naz, Ghulam Abbas Qadri, Rizwan Ali Tunio.

**Validation:** Sarmad Jamal Siddiqi.

**Writing – original draft:** Sarmad Jamal Siddiqi.

**Writing – review & editing:** Rosnah Sutan, Mariam Ashraf, Sadia Ajaib, Zaleha Md Isa, Shabnam Naz, Ghulam Abbas Qadri, Rizwan Ali Tunio.

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
