## [Decision Letter · Decision Letter 0]

23 Nov 2025

PGPH-D-25-02524

Adaptation and Validation of HIV/AIDs module for university students in Larkana, Sindh, Pakistan

Dear Dr. Ashraf,

Thank you for submitting your manuscript to PLOS Global Public Health. After careful consideration, we feel that it has merit but does not fully meet PLOS Global Public Health’s publication criteria as it currently stands. Therefore, we invite you to submit a revised version of the manuscript that addresses the points raised during the review process.

We look forward to receiving your revised manuscript.

Kind regards,

Somayeh Hessam

Academic Editor

Journal Requirements:

Additional Editor Comments (if provided):

Reviewers' comments:

Reviewer's Responses to Questions

**Comments to the Author**

1. Does this manuscript meet PLOS Global Public Health’s publication criteria?

Reviewer #1: Yes

Reviewer #2: Yes

2. Has the statistical analysis been performed appropriately and rigorously?

Reviewer #1: Yes

Reviewer #2: Yes

3. Have the authors made all data underlying the findings in their manuscript fully available (please refer to the Data Availability Statement at the start of the manuscript PDF file)?

Reviewer #1: Yes

Reviewer #2: Yes

4. Is the manuscript presented in an intelligible fashion and written in standard English?

Reviewer #1: Yes

Reviewer #2: Yes

Reviewer #1: I have read the referred article with keen interest. The information is interesting and innovative; conclusion section is interesting and authors can improve it further. I am recommending authors to do a little more work and add latest literate to support the study. The authors need to improve results section. The level of English is good and smooth, e.g., the language standard, specifically the grammar, of sufficient quality to meet scientific merit for publication. However, I suggest authors to double check for language quality. Describe scientific contribution of the study to the existing body of knowledge. I endorse this manuscript after minor revision as suggested. The topic is interesting and worthy of attention. The methodology is adequate and the conclusions are consistent with the reported data. The manuscript can be improved by expanding the references and citing some recently published articles on this topic.

Authors should consider the following recommendations:

- I recommend further improving the references by citing some of these recent studies on the topic:

Anxiety and Social Functioning among People Living with HIV/AIDS (PLWHA) in Pakistan: A Cross-Sectional Study

Opportunities and Challenges of Generative AI in Pakistani Higher Education: A Qualitative Study on Student Perspectives in Learning, Integrity, and Innovation

Assessing the Feasibility of Metacognitive Training for Patients with Schizophrenia in Pakistan: A Randomized Controlled Trial

Feasibility Evaluation of the Listen Protect Connect (LPC) Intervention for School Students in Pakistan: A Cluster Randomized Controlled Trial

A hindrance to proper health care: psychometric development and validation of opiophobia questionnaire among doctors in Pakistan

Reviewer #2: The manuscript addresses an important public health topic and is well-organized, with a clear. The study design, methodology, and analysis are appropriate, and the findings are clearly presented. Overall, the manuscript is of good quality, provides valuable insights, and is acceptable for publication. I have no further comments.

**Do you want your identity to be public for this peer review?** For information about this choice, including consent withdrawal, please see our Privacy Policy

Reviewer #1: No

Reviewer #2: No

---

## [Decision Letter · Decision Letter 1]

3 Feb 2026

PGPH-D-25-02524R1

Adaptation and Validation of HIV/AIDs module for university students in Larkana, Sindh, Pakistan

Dear Dr. Ashraf,

Thank you for submitting your manuscript to PLOS Global Public Health. After careful consideration, we feel that it has merit but does not fully meet PLOS Global Public Health’s publication criteria as it currently stands. Therefore, we invite you to submit a revised version of the manuscript that addresses the points raised during the review process.

We look forward to receiving your revised manuscript.

Kind regards,

Somayeh Hessam

Academic Editor

Journal Requirements:

Additional Editor Comments (if provided):

Reviewers' comments:

Reviewer's Responses to Questions

**Comments to the Author**

Reviewer #1: All comments have been addressed

Reviewer #3: (No Response)

Reviewer #4: All comments have been addressed

Reviewer #5: (No Response)

publication criteria?

Reviewer #1: Yes

Reviewer #3: Yes

Reviewer #4: Yes

Reviewer #5: Partly

3. Has the statistical analysis been performed appropriately and rigorously?

Reviewer #1: Yes

Reviewer #3: Yes

Reviewer #4: Yes

Reviewer #5: No

4. Have the authors made all data underlying the findings in their manuscript fully available (please refer to the Data Availability Statement at the start of the manuscript PDF file)?

Reviewer #1: Yes

Reviewer #3: Yes

Reviewer #4: Yes

Reviewer #5: Yes

5. Is the manuscript presented in an intelligible fashion and written in standard English?

Reviewer #1: Yes

Reviewer #3: Yes

Reviewer #4: Yes

Reviewer #5: Yes

Reviewer #1: NO

Reviewer #3: I appreciate the preparation and the submission of this article. The language use and command is good, and the data and analysis are presented with clarity. My comment revolves around the consistency of the use of terminologies (i.e. the title uses small s for AIDS). I would also suggest reviewing some text and making it more person-centered; i.e. the first part of the introduction could have been written as "60 million individuals in the world are living with HIV." Similarly, authors may consider strengthening the introduction with more evidence from Asia-Pacific, and strengthening the rationale on the decision to select Larkana City in relation to epidemiology and other factors. Authors should also verify the use of citation as some of them are incorrect, i.e. citation 5 that indicates "delayed marriage" as risk factor for increased vulnerability to STI among youngsters but wasn't in the citation when I double-checked.

Reviewer #4: All comments have been addressed. this includes addition of something on physco-social issues. The appropriate references have been added.

Reviewer #5: Adaptation and Validation of HIV/AIDs module for university students in Larkana, Sindh,

Pakistan

General remarks:

I see this manuscript has already undergone some form of refinery. It is well written in a professional and intelligent fashion. However, I have some few concerns:

Concern 1:

Under Phase II of study design and duration, the authors mentioned that ‘The questionnaire was created following the guidelines of Jordan et al.’ The authors forgot to add this in-text citation to their list of references. I strongly suggest that they do so. Again, the authors should insert appropriate in-text citation, according to the journal’s style.

Concern 2:

Was the Ethics approval from Malaysia necessary? I know this is not a multi-center/country study, and I expect the Ethics review committee from Pakistan to grant approval for this work, and not Malaysia.

Concern 3:

What is the rationale behind the selection of 48 students? My search reveals the student population is over 500, so why only 48 students? Is this number not too small to draw any meaningful conclusion? I see the small number was stated as a limitation, but I believe this is not enough, especially in a respected journal like the public health journal. Authors should include sample size calculation in the manuscript, and explain the rationale for arriving at 48. This cannot be a random number.

Concern 4:

Under content validity, the authors wrote:

‘For this purpose, the study calculated the content validation indices recommended by Lynn (1986) and Davis (1992)14, and Polit et al15’.

- There is no Davis (1992) in the authors list of references.

- The reference Lynn (1986) is too old for a modern study of this nature. The reference is 40 years now. Don’t we have any modern (less than 5 or 10 years) study to reference? Perhaps, there are some updates in modern studies that the authors would love to know.

- Reference number 15 has only two authors. Do we apply ‘et al’ to two authors?

Concern 5:

Under Methods, the authors should provide a section/sub-section for Data collection and statistical analysis. It would be interesting to know (in a systematic fashion) how the data collected was analyzed, and what statistical tool(s)/program(s) was used for this analysis.

Concern 6:

What is the full meaning of QUEST? How many students from the engineering section and how many students from the medical section? This part of the study lack clarity. Under the methods, it would be great to see a sub-section talking about ‘study site/area’. The authors can talk a little about the university involved, and why this institution, among many other institutions, was selected for this study, etc.

**Do you want your identity to be public for this peer review?** For information about this choice, including consent withdrawal, please see our Privacy Policy

Reviewer #1: No

Reviewer #3: No

Reviewer #4: **Yes:** Ester Acen

Reviewer #5: No

---

## [Decision Letter · Decision Letter 2]

2 Mar 2026

PGPH-D-25-02524R2

Adaptation and Validation of HIV/AIDs module for university students in Larkana, Sindh, Pakistan

Dear Dr. Ashraf,

Thank you for submitting your manuscript to PLOS Global Public Health. After careful consideration, we feel that it has merit but does not fully meet PLOS Global Public Health’s publication criteria as it currently stands. Therefore, we invite you to submit a revised version of the manuscript that addresses the points raised during the review process.

We look forward to receiving your revised manuscript.

Kind regards,

Somayeh Hessam

Academic Editor

Journal Requirements:

Additional Editor Comments (if provided):

Reviewers' comments:

Reviewer's Responses to Questions

**Comments to the Author**

Reviewer #3: (No Response)

Reviewer #5: All comments have been addressed

publication criteria?

Reviewer #3: Partly

Reviewer #5: Yes

3. Has the statistical analysis been performed appropriately and rigorously?

Reviewer #3: Yes

Reviewer #5: Yes

4. Have the authors made all data underlying the findings in their manuscript fully available (please refer to the Data Availability Statement at the start of the manuscript PDF file)?

Reviewer #3: Yes

Reviewer #5: Yes

5. Is the manuscript presented in an intelligible fashion and written in standard English?

Reviewer #3: Yes

Reviewer #5: Yes

Reviewer #3: There are some statements that need proper citation to ensure it reliability. Likewise, the authors need to double-check whether all citations do reflect the argument that they are presenting. In particular, there is one statement regarding the effect of delayed marriages on the STI vulnerability of young people but the original publication that is being cited has not referred to this at all - I wonder if this is a mistake in terms of citation, and if yes, I suggest looking for proper citation.

Reviewer #5: Authors have addressed all concerns. Thank you.

**Do you want your identity to be public for this peer review?** For information about this choice, including consent withdrawal, please see our Privacy Policy

Reviewer #3: No

Reviewer #5: No

---

## [Decision Letter · Decision Letter 3]

8 Mar 2026

Adaptation and Validation of HIV/AIDs module for university students in Larkana, Sindh, Pakistan

PGPH-D-25-02524R3

Dear Dr Ashraf,

We are pleased to inform you that your manuscript 'Adaptation and Validation of HIV/AIDs module for university students in Larkana, Sindh, Pakistan' has been provisionally accepted for publication in PLOS Global Public Health.

Best regards,

Somayeh Hessam

Academic Editor

Reviewer Comments (if any, and for reference):

Reviewer's Responses to Questions

**Comments to the Author**

Reviewer #3: All comments have been addressed

Reviewer #5: All comments have been addressed

publication criteria?

Reviewer #3: Yes

Reviewer #5: Yes

3. Has the statistical analysis been performed appropriately and rigorously?

Reviewer #3: Yes

Reviewer #5: Yes

4. Have the authors made all data underlying the findings in their manuscript fully available (please refer to the Data Availability Statement at the start of the manuscript PDF file)?

Reviewer #3: Yes

Reviewer #5: Yes

5. Is the manuscript presented in an intelligible fashion and written in standard English?

Reviewer #3: Yes

Reviewer #5: Yes

Reviewer #3: (No Response)

Reviewer #5: All comments have been addresses. I do not have any other concern(s).

**Do you want your identity to be public for this peer review?** For information about this choice, including consent withdrawal, please see our Privacy Policy

Reviewer #3: No

Reviewer #5: No
